# Development and Molecular Cytogenetic Identification of Two Wheat-*Aegilops geniculata* Roth 7M^g^ Chromosome Substitution Lines with Resistance to Fusarium Head Blight, Powdery Mildew and Stripe Rust

**DOI:** 10.3390/ijms23137056

**Published:** 2022-06-24

**Authors:** Xiaoying Yang, Maoru Xu, Yongfu Wang, Xiaofang Cheng, Chenxi Huang, Hong Zhang, Tingdong Li, Changyou Wang, Chunhuan Chen, Yajuan Wang, Wanquan Ji

**Affiliations:** 1College of Agronomy, Northwest A&F University, Yangling 712100, China; yangxiaoying1203@163.com (X.Y.); xmaoru1856@163.com (M.X.); wyf063575@163.com (Y.W.); chengxf611@163.com (X.C.); huangchenxiobst@nwafu.edu.cn (C.H.); zhangh1129@nwafu.edu.cn (H.Z.); tingdongli@nwafu.edu.cn (T.L.); chywang2004@126.com (C.W.); chchch8898@163.com (C.C.); 2State Key Laboratory of Crop Stress Biology for Arid Areas, Yangling 712100, China; 3Shaanxi Research Station of Crop Gene Resources and Germplasm Enhancement, Ministry of Agriculture, Yangling 712100, China

**Keywords:** *Aegilops geniculata* Roth, substitution lines, Fusarium head blight (*Fhb*), powdery mildew, stripe rust

## Abstract

Fusarium head blight (*Fhb*), powdery mildew, and stripe rust are major wheat diseases globally. *Aegilops geniculata* Roth (U^g^U^g^M^g^M^g^, 2*n* = 4x = 28), a wild relative of common wheat, is valuable germplasm of disease resistance for wheat improvement and breeding. Here, we report the development and characterization of two substitution accessions with high resistance to powdery mildew, stripe rust and *Fhb* (W623 and W637) derived from hybrid progenies between *Ae. geniculata* and hexaploid wheat Chinese Spring (CS). Fluorescence in situ hybridization (FISH), Genomic in situ hybridizations (GISH), and sequential FISH-GISH studies indicated that the two substitution lines possess 40 wheat chromosomes and 2 *Ae. geniculata* chromosomes. Furthermore, compared that the wheat addition line parent W166, the 2 alien chromosomes from W623 and W637 belong to the 7M^g^ chromosomes of *Ae. geniculata* via sequential FISH-GISH and molecular marker analysis. Nullisomic-tetrasomic analysis for homoeologous group-7 of wheat and FISH revealed that the common wheat chromosomes 7A and 7B were replaced in W623 and W637, respectively. Consequently, lines W623, in which wheat chromosomes 7A were replaced by a pair of *Ae. geniculata* 7M^g^ chromosomes, and W637, which chromosomes 7B were substituted by chromosomes 7M^g^, with resistance to *Fhb*, powdery mildew, and stripe rust. This study has determined that the chromosome 7M^g^ from *Ae. geniculata* exists genes resistant to *Fhb* and powdery mildew.

## 1. Introduction

Fusarium head blight (*Fhb*), mainly caused by the fungus *Fusarium graminearum* Schwabe [1], is one of the most destructive diseases of wheat, barley (*Hordeum vulgare*), and other-grain cereals in many areas of the world [2]. *Fhb* causes seriousyield losses and grain quality decreases due to Fusarium toxins contaminate cereal products, which are harmful to humans and animals [3]. In recent years, the wheat regions in the middle and lower reaches of Yangtze River and Huang-Huai wheat region in China have been frequently and suffering from *Fhb* due to climate change and straw returning under the wheat- maize rotation system [4]. To date, sources of *Fhb* resistance used wheat cultivars can be traced to few parents [5]. Resistance to *Fhb* in wheat is controlled by polygenes, and symptom expression is often modulated by environmental factors [6]. Currently, hundreds of QTL totally identified in a number of germplasms, which were distributed on all 21 wheat chromosomes [7,8]. Fine mapping of five major-effort QTL identified in common wheat, *Fhb*1 [9,10], *Fhb4* [11], *Fhb5* [12] and *Fhb7* [13] which were widely distributed. Only a few sources of resistance are likely to cause widespread disease epidemics upon that resistance is lost. The development and characterization of new resistance sources will provide broad sources of disease resistance. Therefore, the development and identification of wild relatives of wheat-resistant materials in breeding research are quite meaningful. However, little research has been reported on the *Fhb* resistance of *Ae. geniculata*.

The genomic composition of *Ae. geniculata* is U^g^U^g^M^g^M^g^, in which the U^g^ genome was derived from the *Ae. umbellulate* Zhuk (2*n* = 2x = 14, UU), and the M^g^ genome originated from the *Ae.* comosa Sm. In Sibth. & Sm. (2*n* = 2x = 14, MM) [14]. Numerous researches on wild hybridization of *Ae. geniculata* and common wheat have been reported, and mass valuable genes have been exploited and utilized. Chromosomes 1U^g^ from common wheat-*Ae. geniculata* addition line improves the dough rheological properties [15]. thereby improving the quality of wheat, indicating that the chromosome 1U^g^ can advance crop quality. In addition, there are several genes involved in resistance to various diseases have been transferred from that species into common wheat. In 2007, powdery mildew resistance gene *Pm*29 was derived from the wheat-*Ae. geniculata* disomic addition line Poros wheat [16], and the stem rust resistance gene *Sr53* was derived from the wheat-*Ae. geniculata* 5M^g^ translocation line [17]. Wang reported on the research of *Ae. geniculata* 7M^g^ addition and substitution lines of resistance to powdery mildew [14,18].

In the past few decades, due to the domestication and origin of common wheat along with artificial selection, the genetic diversity of wheat cultivars has been increasingly narrowed [19], which introduction of wild relatives is one of the most important ways for broadening the genetic variation. Distance hybridization is one of the most important ways for broadening the genetic variation and breeding new wheat cultivars. Ciferri et al. [20] reported the first interspecific wheat hybrid experiments in 1806. After that, the wheat breeding of distant hybridization involving wild relatives of common wheat has gradually attracted great attention and developed to a certain extent [21]. The creation of intermediate materials for wheat and wild relatives is a crucial step in wheat chromosome engineering breeding. The wheat-*Thinopyrum ponticum* (2*n* = 10x = 70) partial amphiploids were created in the 1970s via distant hybridization [22], and then, Xiaoyan 6 pedigree-related cultivars were produced [23], which made a huge contribution to Chinese varieties. In this study, we obtained two key materials substitution lines in the breeding process through distant hybridization between common wheat and *Ae. geniculata*, which provided important resources for chromosome engineering breeding.

Powdery mildew, which is a recurring disease caused by *Blumeria graminis* f. sp. *tritici* (*Bgt*) and is one of the most destructive diseases in the wheat-growing areas in the world [24]. Wheat stripe rust (or yellow rust), caused by *Puccinia striiformis* f. sp. *tritici*, is one of the considerable diseases of wheat worldwide [25]. In the meantime, stripe rust and powdery mildew have been widespread in the main wheat-producing regions of China [26]. Due to rapid climate and environmental change, it is really necessary to obtain and utilize new germplasms with resistance genes to develop resistant wheat cultivars [27]. With climate change, the physiological races of the cultivars will be easily overcome by the new races, losing the resistance [28,29], therefore, developing and deploying resistant cultivars is the most economical and safe measure.

In this study, two novel wheat-*Ae. geniculata* 7M^g^ substitution lines W623 and W637 were produced through hybridization of between the Abbondanza nullisomic lines and wheat-*Ae. geniculata* 7M^g^ disomic addition line W166. The researchers used a combination of cytogenetic characteristics, functional molecular markers, in situ hybridizations, and disease resistance to identify the chromosome composition and genetic characteristics of disease resistance of the two substitution lines, which can be used as intermediate materials for wheat chromosome engineering breeding.

## 2. Results

### 2.1. Cytogenetic Analysis of W623 and W637

Cytogenetic analysis revealed that about 97% of metaphase mitotic root tip cells (RTCs) of W623 (Figure 1A) and W637 (Figure 1D) indicated 42 chromosomes from different plants. During the meiotic metaphase and anaphase in PMCs of W623 (Figure 1B) and W637 (Figure 1E) had a chromosome configuration of 2*n* = 21II, univalent, trivalent, or quadrivalent chromosomes were not observed. Meanwhile, all chromosomes were evenly distributed on both sides of the equatorial plate, no chromosome was legged at meiotic anaphase I of two lines (Figure 1C,F). Therefore, all results that line W623 and W637 exhibited highly cytological stability.

### 2.2. Sequential FISH-GISH Analysis

GISH and FISH analyses were performed to examine the chromosome composition of W623 and W637 (Figure 2). When *Ae. geniculata* total genomic DNA was used as the labeled probe and sheared CS genomic DNA as the blocker, we observed 40 wheat chromosomes plus two *Ae. geniculata* chromosomes in both W623 and W637 (Figure 2A,D). To further determine the identities of the wheat chromosomes involved in the substitutions, sequential FISH-GISH analysis was performed on the same slide. FISH analysis was primarily performed using two repetitive oligonucleotide probes, Oligo-pSc119.2 (green) and Oligo-pTa535 (red) (Figure 2B,E), and then GISH analysis was carried out (Figure 2C,F). The sequential FISH-GISH screening of mitotic cell divisions showed that W623 contained 40 chromosomes of wheat, two chromosomes, and lost two 7A chromosomes of wheat (Figure 2B). As well as, the sequential FISH-GISH results indicated that W637 also completed 40 chromosomes of wheat, plus two *Ae. geniculata* chromosomes, but lost two 7B chromosomes of wheat (Figure 2E).

To further determine whether the identity of the alien chromosomes of W623 and W637 were 7M^g^, FISH of the wheat-*Ae. geniculata* 7M^g^ addition line W166 (2*n* = 44) was performed using Oligo-pSc119.2 and Oligo-pTa535 probes, and the special GISH of W623 using was also performed using *Ae. geniculata* total genomic DNA and Oligo-pTa535 as probes (Figure 3). That result showed that the red signal of pTa535 appeared at the end of two 7M^g^ chromosomes arms, which was the same as the FISH pattern of alien chromosomes in W623 and W637.

### 2.3. Molecular Markers Analysis

To further determine the chromosomal composition of the two substitution lines, molecular markers analysis was performed. Two substitution lines were analyzed by employing 72 EST-STS markers and 135 PLUG markers on seven different homologous groups in wheat to clarify the homologous group relationships of the alien chromosomes (Figure 4). In the present study, five PLUG markers (*TNAC1782-TaqI*, *TNAC1845-TaqI, TNAC1929-TaqI/HaeIII, TNAC1888-HaeIII,* and *TNAC1811-HaeIII*) (Table 1) on wheat homoeologous group 7 chromosomes generated *Ae. geniculata* specific bands in four *Ae. geniculata*, the wheat-*Ae. geniculata* addition line W166, W623, and W637. In contrast, no specific fragment from CS, as well as Abbondanza nullisomic line 7A and Abbondanza nullisomic line 7B (Figure 4A–F). This indicated that W623 and W637 had a specific fragment of *Ae. geniculata.*

Nulli-tetrasomic analysis was implemented to analyze the lost pair of chromosomes in W623 and W637 andCS nulli-tetrasomic lines N7AT7B, N7BT7D, and N7DT7A. As shown in Figure 4F,G, two EST-STS markers (*BE426692* and *BE637663*), which could specifically amplify fragments from *Ae. geniculata*, W166, W623, and W637, did not amplify specific bands from other lines, as the white arrow indicated. In addition, compared with the bands of CS and nulli-tetrasomic lines, it was shown that the chromosomes 7A and 7B specific bands were significantly absent in W623 and W637, respectively, as the red arrow indicated. Molecular markers and nulli-tetrasomic analyses showed that the two substitution lines were missing a pair of 7A and 7B chromosomes, respectively, and added a pair of chromosomes 7M^g^ in *Ae. geniculata*.

### 2.4. Resistance against F. graminearum

To determine its function in resistance against *F. graminearum*, two substitution lines W623 and W637 as well as their parents *Ae. geniculata*, wheat-*Ae. geniculata* 7M^g^ addition line W166, and CS were evaluated in the field (Figure 5A and Appendix A). W623, W637, W166, and *Ae. geniculata* showed a lower *Fhb* infected spike ratio due to only one spikelet infection compared with their parent CS, indicating that the resistance gene of two substitution lines to *Fhb* was derived from the ancestral parent *Ae. geniculata*. On the contrary, another ancestral parent, Chinese Spring, was not resistant to *Fhb* at all because all spikelets were infected and died (Figure 5A1).

Because of the dependence of QTL effects on the environment, verification and marker saturation of detected QTL are always necessary for breeding and disease resistance research. To achieve this goal, we used functional markers or linkage markers with *Fhb* resistance gene or QTLs *Fhb1*, *Fhb4, Fhb5,* and *Fhb7* of two lines, and their parents were detected (Figure 6), for example, an *Fhb* gene linked to *Xgwm149* on chromosome 4BS. As shown in Figure 7, *Ae. geniculata* and its derived offspring don’t carry the above Table 2 resistance QTL. In order to validate whether the *GST* gene exists in the wheat background, primers (*F7-1* and *F7-7*) were designed using its specific fragment and its function was clarified. The sequence alignment results are shown in Appendix A. The above results show that the plant material containing the 7M^g^ chromosome has the characteristics of resistance to *Fhb* and does not contain the QTLs *Fhb1*, *Fhb4, Fhb5,* and *Fhb7,* suggesting that there may be a new resistance QTL of *Fhb* in these materials, which was derived from the 7M^g^ chromosome of *Ae. geniculata.*

### 2.5. Powdery Mildew and Stripe Rust Resistance

The reactions to powdery mildew in W623 and W637 were evaluated at the seedling and adult plant stages using the wheat cultivar Shaanyou 225 as a control. At the seedling and the adult plant stages, *Ae. geniculata*, wheat-*Ae. geniculata* 7M^g^ addition line W166, and two substitution lines were immune to the powdery mildew race E09 isolate with an IT score of 0 (Figure 5B and Appendix A). In contrast, the Abbondanza nullisomic line, as well as susceptible control Shaanyou 225, were highly susceptible with an IT score of 4, and CS was moderately susceptible to E09 isolate with an IT score of 3. The results suggested that two substitution lines W623 and W637 had powdery mildew resistance in the whole stage, and these powdery mildew resistance genes were derived from the *Ae. geniculata*.

For testing the stripe rust reaction at the adult stage, W623, W637, and control variety Huixianhong were inoculated a mixture of *Pst* races CYR32, CYR33, and CYR34. In the field test, the stripe rust developed to a sufficient level to judge the disease resistance of the experimental material. Two lines W623 and W637 were nearly immune to stripe rust, meantime, the susceptible control Huixianhong was considered to be susceptible (Figure 5C).

## 3. Discussion

Since the mutual compensation between the alien chromosomes from wild relatives and individual chromosomes of the parents often produce the substitution lines via the distant hybridization [26]. To date, excellent phenotypic traits and genotypes between wheat and wild relatives have been successfully produced in current wheat breeding [30]. Creation of a series of tool materials such as substation lines, addition lines, and translocation lines is a critical step in introducing the excellent genes of the wild relative into cultivated wheat. After that, the exogenous fragments are fragmented by hybridization or radiation to produce the introgression lines applied to production. In this research, in situ hybridization analysis confirmed that W623 and W637 are stable wheat-*Ae. geniculata* substitution lines that may be useful for breeding new wheat varieties with improved disease resistance.

Research shows that *Aegilops geniculata* Roth, as members of the tertiary gene pool of wheat [31], harbors many excellent traits for wheat breeding. So far, *T.aestivum-Ae. geniculata* progeny with excellent traits has been reported in large numbers, involving two genomes of the U^g^ and M^g^ genome. Firebe et al. [32] first carried out the complete set of *T. aestivum-Ae. geniculata* chromosome addition lines in 1999. Since then, the studies of the progeny of *T. aestivum-Ae. geniculata* became detailed and specific. Chromosome 4M^g^ from wheat-*Ae. geniculata* 4M^g^ (4B) substitution line can induce the formation of supernumerary florets. In addition to improving crop quality, *Ae. geniculata* has been widely studied in carrying disease resistance genes. Stoilova et al. [33] reported that chromosome 6U from wheat- *Ae. geniculata* addition line carrying powdery mildew resistance. The wheat-*Ae. geniculata* introgression T5DL 5DS-5M^g^S (0.95), with stripe rust resistance gene *Yr40* and leaf rust resistance gene *Lr57*, is a source of resistance in Kansas and India [34]. For all that, although many pieces of researches on disease resistance have been reported, there is no report on the study of FHB in offspring from distant hybridization between the common wheat and *Ae. geniculata*. In this study, we develop and identify two wheat- *Ae. geniculata* 7M^g^ (7A) and 7M^g^ (7B) substitution lines carrying disease resistance to powdery mildew and FHB. Compared with previous studies, we believed that chromosomes 7M^g^ from *Ae. geniculata* not only have disease resistance to powdery mildew but also possess good resistance to Fusarium head blight. At the same time, the *Ae. geniculata* substitution line W623 and the disomic substitution line W637, both of which are highly resistant to stripe rust at the adult stage, since no identification of yellow rust at the adult stage was carried out on their parents, the sources of resistance to stripe rust was not clear. We do not know whether the stripe rust accompanied with the 7M^g^ substitution lines was related to the M^g^ genome from wheat-*Ae. geniculata.* However, it is undeniable that two substitution lines have high-level resistance to the three main diseases of wheat at the same time, which are rare bridge materials in chromosome engineering breeding.

*Fusarium graminearum* is a caused agent of Fusarium head blight (*Fhb*) [35] and produces the mycotoxin deoxynivalenol (DON) in grain, which is harmful to human and animal health [36]. It is well known that there are very few sources of parents of *Fhb* resistance in wheat breeding, mainly including Sumai 3 and its derivatives, Wangshuibai and Wuhan 1 [8,26]. Development and identification of *Fhb* resistance genes of wheat relatives and introduction of resistance genes to common wheat by chromosome engineering breeding are important methods to solve the shortage of resistance materials. Zhao et al. [37] reported that a wheat-*Leymus mollis* (NsNsXmXm, 2*n* = 4x = 28) double substitution line with resistance to yellow rust and *Fhb*. Wheat-*Leymus racemosus* alien introgression lines with *Fhb* resistance were described in 2008 [38]. Two wheat-*Elymus repens* (StStStStHH, 2*n* = 6x = 42) lines have determined that the chromosome 3St of *E. repens* harbors gene(s) highly resistant to *Fhb* [39]. And *Fhb7* is a QTL introduced from T*hinopyrum elongatum* (EE, 2*n* = 14) was reported in 2020 [13]. However, few studies have been conducted on the pedigrees of the wheat-*Ae. geniculata.* Herein, we confirmed the resistance of *Ae. geniculata*, the wheat-*Ae. geniculata* 7M^g^ addition line, the wheat- *Ae. geniculata* 7M^g^ (7A) substitution line, and wheat-*Ae. geniculata* 7M^g^ (7B) disomic substitution line to *Fhb*. In consequence, we determined the resistance of wild relatives of *Aegilops geniculata* Roth, and enriched the sources of resistance to *Fhb*, and also identified the resistance of the 7M^g^ chromosome of *Ae. geniculata.*

Distant hybridization is important to develop new excellent germplasm and broaden the genetic base of cultivar wheat. In early studies, the substitution lines, translocation lines, addition lines, and introgression lines in research and breeding are generally produced by distant hybridization between the hexaploid wheat and wild relatives by crop breeders [32,40]. Compared with the substitution and addition lines are more easily to be used in breeding because they carry fewer alien chromosome fragments. To produce wheat-*Ae. geniculata* substitution lines of homologous group-7 in wheat, we used the wheat deletion lines of group-7 and the addition 7M^g^ wheat-*Ae. geniculata* to cross. In the initial stage of this study, we used the wheat-*Ae. geniculata* 7M^g^ disomic addition line W166 and the Abbondanza nullisomic lines of (7A and 7B) for hybridization, and purposefully substituted a pair of common wheat 7A and 7B chromosomes in the addition line to produce the substitution lines. In this process, we used cytological and disease resistance to powdery mildew analysis to select the substitution lines of the hybridization between the deletion lines and W166. To explore whether there were other excellent traits in W623 and W637, we inoculated them with the race PH-1, and we finally determined that these two substitution lines have resistance to *Fhb*.

## 4. Materials and Methods

### 4.1. Plant Materials and Development of W623, W637

To generate the substitution lines, the crossing strategy was carried out as illustrated in Figure 1. Chinese Spring (CS) and *Ae. geniculata* were used as an early parent in this study. Lines W623 and W637 were selected from a progeny of a cross between Abbondanza nullisomic line 7A (genome composition: 12A + 14B + 14D, 2*n* = 40) and wheat-*Ae. geniculata* 7M^g^ addition line W166 (NA0973-5-4-1-2-9-1, AABBDD, 2*n* = 44) [14], and the Abbondanza nullisomic line 7B (genome composition:14A + 12B + 14D, 2*n* = 40) and the line W166, respectively. The wheat cultivars Shanyou 225, Huixianhong and Aikang 58 were used as the susceptible control of powdery mildew, stripe rust and FHB. The above-mentioned materials were planted at Yangling, Shaanxi Province, China (108°4′ E, 34°16′ N) during the 2018–2021 growing season.

The wheat-*Ae. geniculata* 7M^g^ disomic addition line W166, which has the characteristics of more kernels per spikelet, and higher thousand-kernel weight, as well as disease resistance to powdery mildew, was originally developed and identified from the crosses CS/*Ae. geniculata*//CS BC_1_F_6_ progenies [14]. To produce the wheat-*Ae. geniculata* 7M^g^ substitution lines, we crossed W166 with the Abbondanza nullisomic line 7A and Abbondanza nullisomic line 7B, and F_1_ was backcrossed with Abbondanza nullisomic line 7A and Abbondanza nullisomic line 7B, respectively. The crossing strategy was carried out as illustrated in Figure 1. 

Then, identified pollen mother cell chromosomal configurations of F_1_ derived lines from between the addition line W166 and Abbondanza nullisomic line 7A and Abbondanza nullisomic line 7B, and screened out that plants for harvest and seed retention of the PMCs are 2*n* = 42 = 20II + I + I (Figure 7A) and resistant to powdery mildew. The seeds selected from F_1_ plants were bulked and advanced to the next generation. The next year, the pollen mother cell chromosomal configurations of BC_1_F_1_ interspecific hybrids were identified and investigated into the number and the configurations of the PMCs are 2*n* = 41 = 20II + I (Figure 7B) and also resistant to powdery mildew for continuous self-crossing until producing stable substitution lines. Two stable disomic substitution lines W623 and W637, with high resistance to powdery mildew over 2 years of observation, were isolated from 65 W166/2* Abbondanza nullisomic line 7A or Abbondanza nullisomic line 7B BC_1_F_4_ progeny (Figure 7).

### 4.2. Cytological Identification

When the flag leaf of wheat was spread from late Marth to early April, the young spikes were respectively collected and fixed in Carnoy’s solution to perform meiosis analysis, which was treated as described previously [41]. Fresh root tips were sampled from germinating seeds treated with nitrous oxide (N_2_O) for 2 h and then fixed in 90% glacial acid and stored in 70% *v*/*v* ethanol to perform mitosis analysis and chromosome slides preparation [40,42]. The pollen mother cells (PMCs) and the root tip cells (RTCs) were analyzed and photographed with an Olympus BX-43 microscope (Japan).

### 4.3. GISH and Sequential FISH-GISH

Genomic DNAs were isolated from *Ae. geniculata* and CS from fresh leaves by a modified CTAB method [43], then *Ae. geniculata* DNA was labeled fluorescein-12-dUTP (green) by the nick translation method [44,45] and used as a probe. Sheared genomic DNA of CS was used as blocking DNA. GISH analysis was performed according to Han et al. [44] and Gong et al. [39]. Probe labeling and FISH-GISH were performed as described in Tang et al. [46].

### 4.4. Molecular Markers Analysis and Nulli-Tetrasomic Analysis

DNA was extracted from fresh leaves of W166 [14], W623, W627, Abbondanza nullisomic line 7A, Abbondanza nullisomic line 7B, and CS nulli-tetrasomic lines N7AT7B, N7BT7D, and N7DT7A, using the modified CTAB method, then used as the molecular marker analysis and nulli-tetrasomic analysis. The polymerase chain reaction (PCR)-based landmark unique gene (PLUG) [47] primer pairs were used to determine the alien chromatin of W623 and W627. The expressed sequence tagged sequence site (EST-STS) markers were used to identify the wheat chromatin in both lines. All PCR markers distributed in seven homoeologous groups of wheat were obtained from the Wheat Haplotype Polymorphisms website (https://wheat.pw.usda.gov/ accessed on 12 May 2020), and the details are shown in Table 1. The PCR amplification was conducted as previously described by Wu et al. [48].

### 4.5. Maintenance and Preparation of Inoculum

The *F. graminearum* inoculum used in all experiments was “PH-1 (NRRL 31084)”, a major pathogen of cultivated cereals and has completed genome sequencing [49], was kindly provided by Cong Jiang (Northwest A&F University, Yangling, China). Conidia were cultivated in mung bean (MB) liquid medium, and then conidia were harvested from 5-day-old at 25 °C MB and re-suspended to 2.5 × 10^5^ spores per ml of PH-1between the lemma and palea in sterile distilled water (SDW) by blood counting chamber. The cultural conditions and fungal transformation were followed by Jiang et al. [50]. When the fluffy stigma and ovary appear in flowering heads of plants were inoculated with 10 μL of conidium suspensions at the spikelet. The plant inoculation experiments were performed using single-floret inoculation [26]. Inoculated plants were sprayed with and covered for 48 h with a plastic bag to maintain high humidity. Inoculated wheat heads were estimated for diseased spikelets at 18 or 21 dpi to estimate the disease index (number of diseased spikelets per head) [51], in which percentage of diseased spikelets (PDS) was used to measure *Fhb* resistance. Here PDS of plant material was calculated by dividing the number of spikelets with visible disease symptoms 21 days after the inoculation by the total number of spikelets in these spikes. The mean of the disease index was estimated with data from three independent replicates with at least 10 randomly chosen wheat heads examined in each replicate.

To detect the distribution of *Ae. geniculata* and two substitution lines in five major-effect QTL in common wheat, *Fhb1* [9,10], *Fhb4* [11], *Fhb*5 [12] and *Fhb*7 [13], several primers were used and optimized to determine the resistance gene for *Fhb*. Sumai 3 and Wangshuibai were the source of *Fhb* resistance in the population, which segregated for three known *Fhb* resistance QTL on *Fhb1*, *Fhb4* and *Fhb5*. The plants were genotyped using microsatellite markers on chromosome 3BS, 4BL, and 5AS to facilitate selection of *Ae. geniculata* and their derived offspring for QTL intervals on 3BS, 4BL, and 5AS carrying the *Fhb* resistance gene (Table 2). *TaHRC-STS* [52] was known to deletion mutation in *Fhb1* and validated in different types of populations, for the *TaHRC-STS* marker. The fragment of *GST* (*Fhb7*) (Gene ID: Tel7E01T1020600.1 https://ngdc.cncb.ac.cn/, accessed on 13 January 2021) on 7EL chromosome was amplified. The fragments amplified from *Th. Elongatum* (syn. *Agropyrom Elongatum* or *Lophopyrum elongatum*) was clone onto the pMD 19-T Vector (TaKaRa Biotech Co., Beijing, China) for sequencing. Thermal cycling included: 94 °C-3 min, 35 cycles of 94 °C-30 s, 56 °C-1 min, 72 °C-7 min. The sequence comparison analysis was carried out using the software DNAMAN.

**Table 2 ijms-23-07056-t002:** Primers used to amplify four *Fhb* genes.

Marker	QTL	Primer Sequence (5′-3′)	Chr.	Tm (°C)	Reference
*TaHRC-STS*	*Fhb1*	F: ATTCCTACTAGCCGCCTGGT R: ACTGGGGCAAGCAAACATTG	3BS	65	Su et al. [52]
*Xgwm 149*	*Fhb4*	F: CATTGTTTTCTGCCTCTAGCC R: CTAGCATCGAACCTGAACAAG	4BS	56	Xue et al. [11]
*Xgwm 304*	*Fhb5*	F: AGGAAACAGAAATATCGCGG R: AGGACTGTGGGGAATGAATG	5AL	56	Xue et al. [12]
*F7-1*	*Fhb7*	F: AGACTGGCCCTCAACTTCAA R: CGACAATCATGTCCGCATAC	7EL	56	At this article
*F7-7*	*Fhb7*	F: GATGCAGTCCCTCCGAAACA R: ACCGACAATCATGTCCGCAT	7EL	55	At this article

### 4.6. Disease Resistance of Powdery Mildew and Stripe Rust

The powdery mildew resistance of plants was evaluated at seedling and adult stages. The wheat variety Shanyou 225 was employed for inoculation of *Blumeria graminis* f. sp. *tritici* (*Bgt*) pathotype E09 in a plant incubator and field. Powdery mildew reactions of two lines and their parents to *Bgt* race E09 at the seeding stage were assessed at green house. At 14 days after the initial inoculated, when the pustules were fully developed on Shanyou 225, the percentage of the powdery mildew spores covered the total area of the leaves at the same position on each plant were recorded by the plant responses. Powdery mildew infection type (IT) from 0 to 4 was identified as described by Jorgensen et al. [24]. The plants were evaluated and IT were recorded according with IT 0–2 were considered resistant, while those with IT 3–4 susceptible. The reactions to powdery mildew at the adult stage were tested on two substitution lines, addition line W166, their parents and susceptible cultivar Shanyou 225 during the 2018–2021 growing season at Yangling, Shaanxi Province, China. After wheat heading, when the susceptible control Shanyou 225 showed disease presentation. Powdery mildew disease reaction was were evaluated and IT recorded according to a 0–9 scale, which an IT of 0–4 was considered as resistant and an IT of 5–9 indicated susceptibility [53].

To assess the reactions of two lines to stripe rust at the adult stage, the plants were inoculated with a mixture of *Pst* CYR32, CYR33, and CYR34, as well as the wheat cultivar Huixianhong was used as the susceptible control, which are prevalent in China as used for artificial inoculation in field in early spring at Northwest A&F University, Yangling, China. When inoculating, spray the mixed strain containing diatomaceous earth on the susceptible material Huixianhong moistened with water in advance, and cover with a plastic bag to moisturize until the plastic bag is removed the next morning. After wheat heading, when the susceptible control Huixianhong showed disease presentation, the plant inoculation experiments and resistance evaluation were performed by Wu et al. [54]. In brief, wheat leaves responses to infection were scored using IT in a 0–4 scale, in which IT = 0, nearly immune; 1, highly resistant; 2, moderately resistant; 3, moderately susceptible and 4, susceptible.

## 5. Conclusions

In this study, we developed and charactered two stable wheat-*Ae. geniculata* 7M^g^ (7A) and 7M^g^ (7B) substitution lines W623 and W637 by cytogenetic analysis, sequential FISH-GISH, molecular markers, and disease resistance to powdery mildew, stripe rust and *Fhb*. Therefore, two substitution lines have the potential to serve as primary material for wheat genetic improvements. Further work is needed to transfer the excellent genes in the chromosomes of foreign genes to common wheat by hybridization or radiation, to identify the genomic composition of these lines by FISH and to evaluate their agronomic performance. In conclusion, the purpose is to transfer the disease resistance gene to common wheat for auxiliary breeding and shorten the breeding period of wheat. Thereby extending the genetic source of wheat breeding, enriching genetic diversity and improving the yield and quality of wheat.

## Figures and Tables

**Figure 1 ijms-23-07056-f001:**
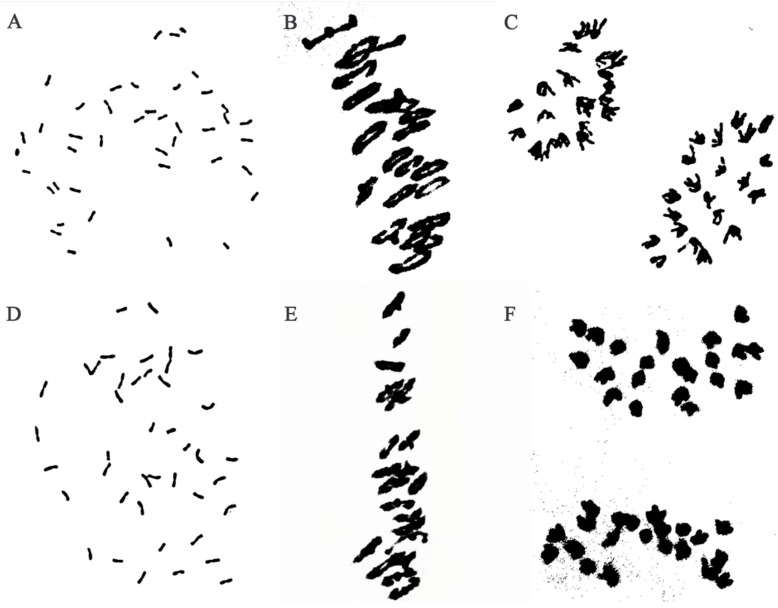
Cytogenetic analysis of W623 and W637. Analysis of root tip cells in W623 (**A**) and W637 (**D**) at mitotic metaphase: 2*n* = 42. Pollen mother cell chromosomal configurations of W623 (**B**) and W637 (**E**) at meiotic metaphase I: 2*n* = 21II. Pollen mother cell chromosomal configurations of W623 (**C**) and W637 (**F**) at anaphase I: 2*n* = 21I + 21I.

**Figure 2 ijms-23-07056-f002:**
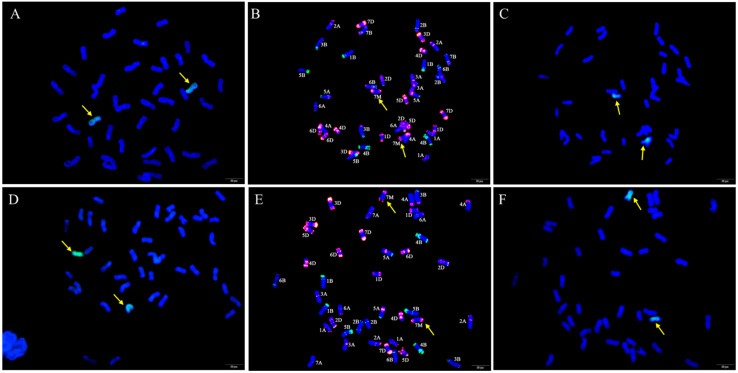
GISH and sequential FISH-GISH analyses of W623 and W637. GISH of a mitotic metaphase cell of W623 (**A**) and W637 (**D**) *Ae. geniculata* genomic DNA (green) as probe and CS genomic DNA as a blocker. FISH of the mitotic metaphase cells of W623 (**B**) and W637 (**E**) using Oligo-pSc119.2 (green) and Oligo-pTa535 (red) were used as probes. GISH in sequential FISH-GISH of W623 (**C**) and W637 (**F**). The yellow arrows indicate *Ae. geniculata* chromosomes in W623 and W637.

**Figure 3 ijms-23-07056-f003:**
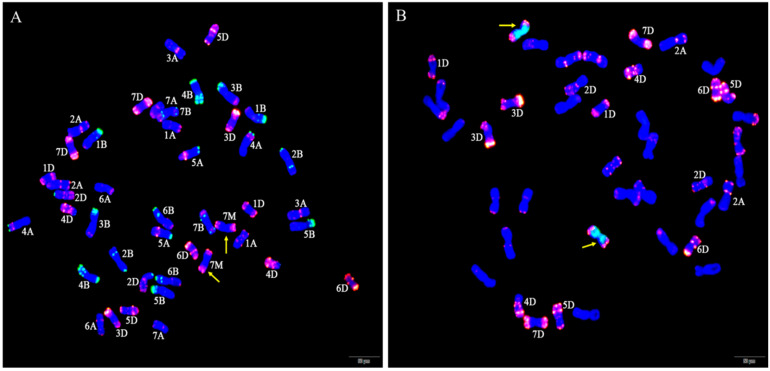
FISH and special GISH analysis. (**A**) Oligo-pSc119.2 (green) and Oligo-pTa535 (red) as probes used in W166. (**B**) *Ae. geniculata* genomic DNA (green), Oligo-pTa535 (red) as probes, and shared CS genomic DNA as a blocker in W623. The yellow arrows indicate *Ae. geniculata* chromosomes.

**Figure 4 ijms-23-07056-f004:**
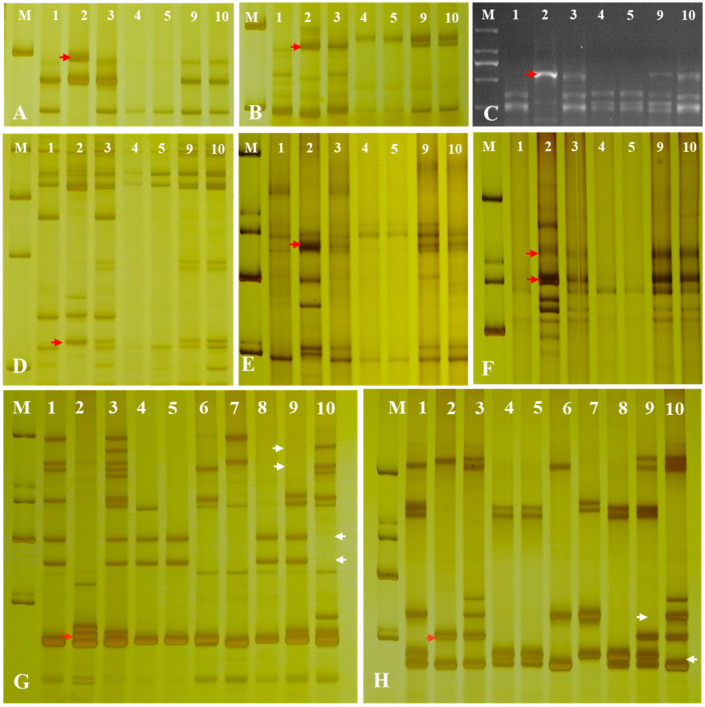
Molecular markers analysis of W623 and W637. M, marker; 1, *Triticum aestivum* cv. Chinese Spring; 2, *Ae. geniculata*; 3, W166; 4, Abbondanza nullisomic line 7A; Lane 5, Abbondanza nullisomic line 7B; 6, CSN7AT7D; 7, CSN7DT7B; 8, CSN7BT7A; 9, W623; 10, W637; (**A**) TANC1782-Taq*I*. (**B**) TANC1845-Taq*I*. (**C**) TANC1811-*HaeIII*. (**D**) TANC1888- *HaeIII*. (**E**) TANC1929-Taq*I*. (**F**) TANC1929- *HaeIII*. (**G**) *BE426692*. (**H**) *BE637663*. The red arrow indicated an *Ae. geniculata*-specific bands and the white arrow directed CS and the nulli-tetrasomic specific bands.

**Figure 5 ijms-23-07056-f005:**
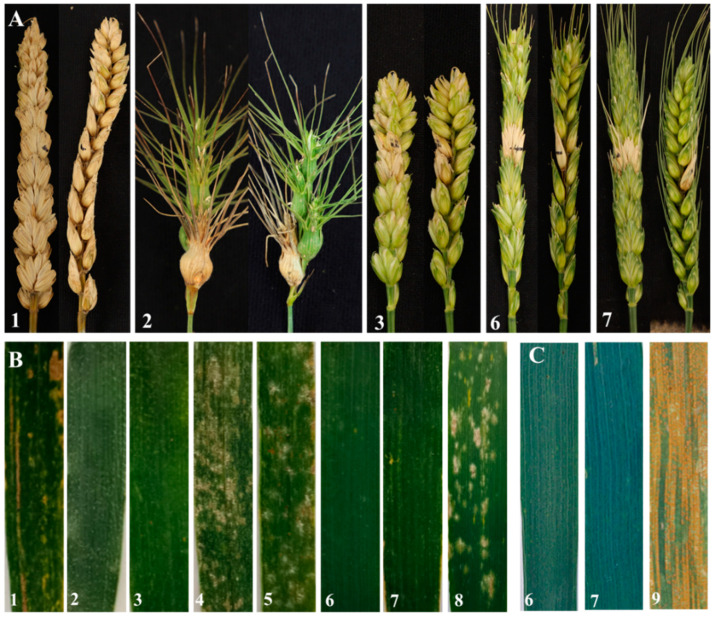
Disease reaction of two lines and their parents. (**A**) Spikes infected with *Fusarium graminearum*. (**B**) Leaves infected with *Blumeria graminis* at the adult stage. (**C**) Leaves infected with *Puccinia striiformis* at the adult stage. 1, *Triticum aestivum* cv. Chinese Spring; 2, *Ae. geniculata*; 3, wheat-*Ae. geniculata* 7M^g^ addition line W166; 4, Abbondanza nullisomic line 7A; 5, Abbondanza nullisomic line 7B; 6, W623; 7, W637; 8, Shaanyou 225; 9, Huixianhong.

**Figure 6 ijms-23-07056-f006:**
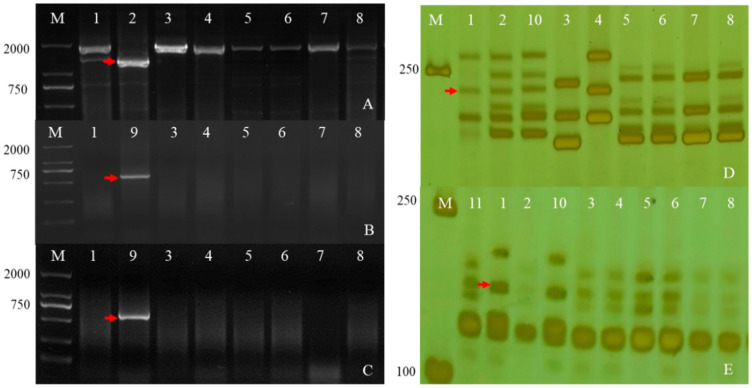
Detection *Fhb1* (**A**), *Fhb7* (**B**,**C**), *Fhb5* (**D**) and *Fhb4* (**E**) using the markers *TaHRC-STS*, *F7-1*, *F7-7*, *Xgwm304* and *Xgwm149*. The red arrows indicate Fhb-specific bands. (M). DL2000. (1). CS. (2). Sumai3. (3). SY159. (4). W166. (5). Abbondanza nullisomic line 7A. (6). Abbondanza nullisomic line 7B. (7). W623. (8). W637. (9). *Th. Elongatum.* (10). Wangshuibai. (11). Aikang58. (a Chinese wheat cultivar).

**Figure 7 ijms-23-07056-f007:**
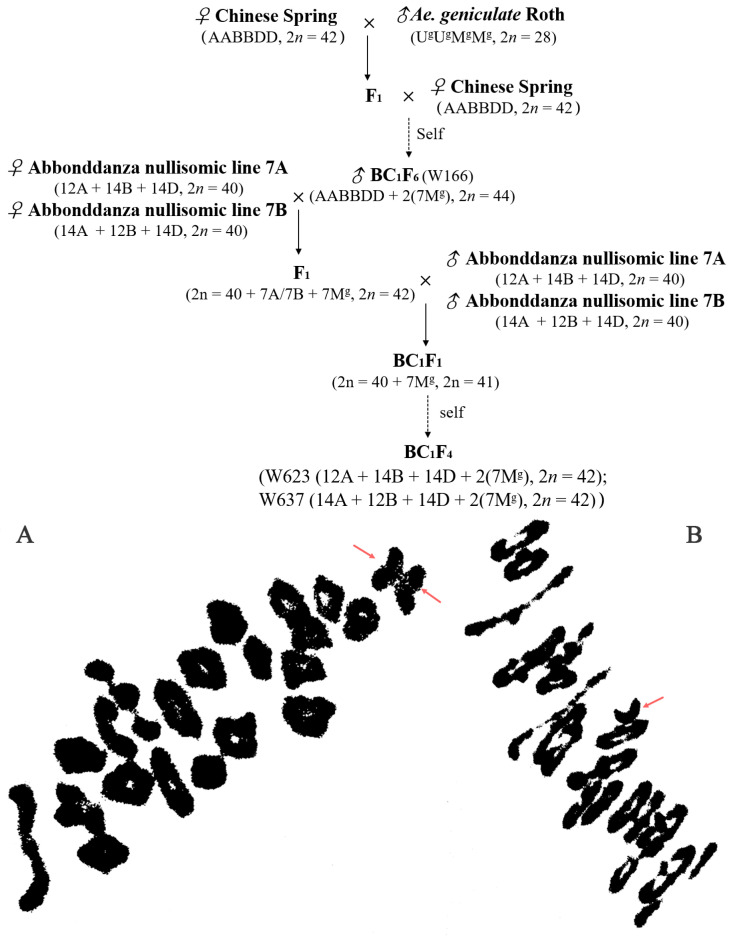
A crossing program showing strategies employed to generate wheat-*Ae. geniculata* substitution lines and cytogenetic analysis of interspecific hybrids. (**A**). pollen mother cell chromosomal configurations of F_1_ originated from the crossing between Abbondanza nullisomic lines and W166 at meiotic metaphase I: 2*n* = 20II + I + I (**B**). pollen mother cell chromosomal configurations of BC_1_F_1_ originated from the crossing between F_1_ and Abbondanza nullisomic lines at meiotic metaphase I: 2*n* = 20II + I. The red arrow indicated univalent chromosomes.

**Table 1 ijms-23-07056-t001:** EST-STS and PLUG markers used in this study of W623 and W637.

Marker	Type	Primer Sequence (5′-3′)	Location	Geltype/Restrictionenzyme	Tm (°C)
*BE637663*	EST-STS	F: ACTGTTGCTTCGCTCCAAGT R: GTTCCATTTCCGATGTGCTC	7AL 7BL 7DL	8% non-denaturing polyacrylamide gel/-	60
*BE426692*	EST-STS	F: CAGAACGAGGACTACCGCTC R: CCAGTAGGTGCCCATCTTGT	7AS 7BS 7DS	8% non-denaturing polyacrylamide gel/-	62
*TNAC1782*	PLUG	F:TCACTGAACAGCCTAGACATGG R: ATTCGCAGACCGCATCTATC	7AS 7BS 7DS	8% non-denaturing polyacrylamide gel/*TaqI*	60
*TNAC1845*	PLUG	F: AATGAACAGCTTGCTTTCTGC R: CAGATGCTCTGGATTTCATGG	7AL 7BL 7DL	8% non-denaturing polyacrylamide gel/*TaqI*	60
*TANC1929*	PLUG	F: GCACCAGAAGGTTCAGTAGCA R: ATCTGTCAGCAGGGCACACT	7AS 7BS 7DS	8% non-denaturing polyacrylamide gel/*TaqI/HaeIII*	60
*TNAC1888*	PLUG	F: AGGGATGTGTTGGAGCTGTTA R: CACAGTGACCTTCTGCTCCTT	7AL 7BL 7DL	8% non-denaturing polyacrylamide gel/*HaeIII*	60
*TNAC1811*	PLUG	F: CTGCTCAACGAGTTCATCGAC R: TTGGAGTGGACGTTGCATT	7AL 7BL 7DL	1.5% agarose gel/*HaeIII*	60

## Data Availability

Not applicable.

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
