# Peer review of "Development and Molecular Cytogenetic Identification of Two Wheat-Aegilops geniculata Roth 7Mg Chromosome Substitution Lines with Resistance to Fusarium Head Blight, Powdery Mildew and Stripe Rust"

_ijms, 2022, doi:10.3390/ijms23137056_

Round 1
Reviewer 1 Report
I recommend the publication of the article because the scientific experimentation concerning the genetic improvement of plants to increase resistance to diseases is of particular interest.
The purpose and goals of the article have been stated and are very interesting. The use of genetic improvement for defense of plants is an important topic especially for the reduction of synthetic products in agriculture. The work done is definitely of international interest and the format applied is definitely suitable for a research article. The work done is original, of particular interest and can definitely stimulate research on this topic. The length of the article is fine for the journal and the graphs and tables are clear and easy to understand. The conclusion summarizes the goals of the work and future prospects.
Author Response
Thank you for your replies and comments concerning our manuscript entitled “Development and molecular cytogenetic identification of two wheat-Aegilops geniculata Roth 7Mg chromosome substitution lines with resistance to Fusarium head blight, powdery mildew and stripe rust” These comments were all valuable and very helpful for revising and improving our paper, as well as providing important guidance for our future research.
Reviewer 2 Report
This manuscript entitled 'Development and molecular cytogenetic identification of two wheat-Aegilops geniculata Roth 7Mg chromosome substitution lines with resistance to Fusarium head blight, powdery mildew and stripe rust’ adds good value to our knowledge and ultimately helps to improve wheat resistance to diseases. However, I have the following concerns before it is accepted for publication.
The authors studied the development and molecular cytogenetic identification of 2 wheat Ae geniculata s. lines resistant to FHB, PM, and Sripe rust. I see the authors detailed sufficient information on FHB but not the other 2 diseases? I suggest to describe more details about the other pathogens or at least to cite other people who did more work on those diseases in terms of inoculation, symptoms, etc.. also, I suggest indicating their impact on wheat production in your country vrs global estimation.
Figures with high quality needed especially Fig 1
First section in the results is purely materials and methods
Use development for new lines rather than creation or generation
Conclusion can be better improved to describe how the outcomes of this study can improve wheat production/resistance to diseases? what is the next step?
Scientific language must be improved
English should be checked well and authors should look at the typos, punctuations etc
Author Response
Thank you for your replies and comments concerning our manuscript entitled “Development and molecular cytogenetic identification of two wheat-Aegilops geniculata Roth 7Mg chromosome substitution lines with resistance to Fusarium head blight, powdery mildew and stripe rust” These comments were all valuable and very helpful for revising and improving our paper, as well as providing important guidance for our future research. We have studied the comments carefully and made appropriate corrections, which we hope meet with your approval.
Language revisions of the entire manuscript can be seen in the manuscript with track changes from PhD Zhang Hong and PhD Li Tingdong, author and acknowledgements are noted. The revised sections can be seen in the manuscript with track changes. The main corrections in the paper and the responses to the reviewers’ comments are in Word in the next section.
Response to Reviewer
- The reviewer suggested to describe more details about the other pathogens or at least to cite other people who did more work on those diseases in terms of inoculation, symptoms, etc..
Re: We accept this comment. More details of powdery mildew and stripe rust, including infection, symptoms and evaluation methods, etc. have been added in section 4.6 of the manuscript.
- The reviewer suggested indicating their impact on wheat production in your country vrs global estimation.
Re: We accept this comment. The content of this part has been supplemented in the introduction.
- Figures with high quality needed especially Fig 1 and First section in the results is purely materials and methods
Re: We accept this comment.
Based on the reviewer's suggestion, we adjusted the first section in the results to the materials and methods, and merged it with the first part, so Figure 1 was adjusted to Figure 7, and the other pictures were moved up in order.
Figures in the manuscript have been revised, including Figures 1, 2, 3, 4, 5, and 7. Figures 1, 2, 3, and 7 provide high-quality pictures on the basis of the previous ones, and the font in the above figures has been modified to "Times New Roman".
- Use development for new lines rather than creation or generation
Re: We accept this comment. We have made changes to this. In the materials and methods section, 4.1.
- Conclusion can be better improved to describe how the outcomes of this study can improve wheat production/resistance to diseases? what is the next step?
Re: We accept this comment. The content of this part has been modified in the conclusion part. It is mainly supplemented that the two substitution lines can be used as intermediate materials for breeding by means of hybridization or radiation, that is, the disease resistance genes can be introduced into the wheat background for production.
- English should be checked well and authors should look at the typos, punctuations etc
Re: We accept this comment. We have made changes to the language section.
Round 2
Reviewer 2 Report
The authors have made significant improvement to the manuscript and therefore i recommend publishing it in the IJMS.